# Impact of a multi-pronged cholera intervention in an endemic setting

**Alexandre Blake**[1]*, **Adam Walder**[2], **Ephraim M. Hanks**[2], **Placide Okitayemba Welo**[3], **Francisco Luquero**[4], **Didier Bompangue**[3,5,6], **Nita Bharti**[1]

1 Biology Department, Center for Infectious Disease Dynamics, Penn State University, University Park, Pennsylvania, United States of America, 2 Statistics Department, Center for Infectious Disease Dynamics, Penn State University, University Park, Pennsylvania, United States of America, 3 Programme National d'Elimination du Choléra et de Lutte contre les autres Maladies Diarrhéiques, Kinshasa, Democratic Republic of the Congo, 4 Epicentre, Paris, France, 5 Department of Ecology and Control of Infectious Diseases, Faculty of Medicine, University of Kinshasa, Kinshasa, Democratic Republic of the Congo, 6 One Health Institute for Africa, University of Kinshasa, Kinshasa, Democratic Republic of the Congo

* agb163@psu.edu

## Abstract

Cholera is a bacterial water-borne diarrheal disease transmitted via the fecal-oral route that causes high morbidity in sub-Saharan Africa and Asia. It is preventable with vaccination, and Water, Sanitation, and Hygiene (WASH) improvements. However, the impact of vaccination in endemic settings remains unclear. Cholera is endemic in the city of Kalemie, on the shore of Lake Tanganyika, in the Democratic Republic of Congo, where both seasonal mobility and the lake, a potential environmental reservoir, may promote transmission. Kalemie received a vaccination campaign and WASH improvements in 2013–2016. We assessed the impact of this intervention to inform future control strategies in endemic settings. We fit compartmental models considering seasonal mobility and environmentally-based transmission. We estimated the number of cases the intervention avoided, and the relative contributions of the elements promoting local cholera transmission. We estimated the intervention avoided 5,259 cases (95% credible interval: 1,576.6–11,337.8) over 118 weeks. Transmission did not rely on seasonal mobility and was primarily environmentally-driven. Removing environmental exposure or contamination could control local transmission. Repeated environmental exposure could maintain high population immunity and decrease the impact of vaccination in similar endemic areas. Addressing environmental exposure and contamination should be the primary target of interventions in such settings.

## Author summary

Cholera is a major global health concern that causes high morbidity. It is a bacterial water-borne disease that can be transmitted via the fecal-oral route or the ingestion of contaminated water. Hence, both population mobility and environmental exposure can promote cholera persistence. The primary tools to prevent cholera include vaccination and Water, Sanitation, and Hygiene (WASH) improvements. The effectiveness of these interventions is well understood in epidemic settings, but their impact in endemic settings

**Data Availability Statement:** The code necessary to reproduce the analysis and the figures is available on https://github.com/bhartilab/cholera_kalemie The data on weekly aggregated number of suspected cholera cases is owned by Doctors

Without Borders (Medecins Sans Frontieres, MSF)-Epicentre. It includes the number of suspected cholera cases residing in Kalemie by ISO week. MSF is an independent, non-profit, international medical humanitarian organization that delivers emergency aid to people affected by armed conflict, epidemics, natural or man-made disasters, or exclusion from health care. Epicentre is an epidemiology, medical research, innovation and training group embedded in MSF. The authors did not receive special privileges in accessing the data. Data can be requested by contacting Klaudia Porten (Klaudia.PORTEN@epicentre.msf.org).

**Funding:** This study was supported by the joint National Institutes of Health (NIH) – National Science Foundation (NSF) - National Institute of Food and Agriculture (NIFA) Ecology and Evolution of Infectious Disease (award R01TW012434 to NB), NSF DEB RAPID (award 2202872 to NB), and NSF DMS (award 2015273 to EH). Funders had no role in the study design, data collection and analysis, decision to publish, or preparation of the manuscript.

**Competing interests:** The authors have declared that no competing interests exist.

is unclear. Achieving cholera elimination requires disentangling the contributors to transmission, specifically population mobility and aquatic reservoirs, and assessing the impact of interventions performed in endemic settings.

This study focuses on Kalemie, a cholera endemic city in the Democratic Republic of Congo, on shore of a lake that serves as a potential environmental reservoir. It quantifies the short-term impact of an intervention that used targeted vaccination and WASH. The study shows that the impact of vaccination was dampened by very high background immunity due to constant environmental exposure. This suggests that WASH improvements should be the primary intervention in such settings despite the time- and resource-intensive nature of implementation.

## Introduction

Cholera is a bacterial water-borne diarrheal disease transmitted through the fecal-oral route. Since the beginning of the 7th cholera pandemic, cholera has been endemic in sub-Saharan Africa (SSA) [1] which now experiences the highest morbidity and mortality globally [2], excluding major epidemic events that occurred in Haiti and Yemen. Typical cholera symptoms include vomiting and diarrhea with rice-water stools, potentially leading to severe dehydration. Individual symptoms can range from asymptomatic infections, to mild infections with symptoms that are hardly distinguishable from other diarrheal diseases, to the typical severe watery diarrhea [3]. The case fatality rate (CFR) can reach 70% among severe cases without appropriate treatment, mainly rehydration [4]. As many as 80% of infections can be asymptomatic in endemic areas [4], resulting in underestimates of cholera burden.

Cholera's causal agent, *Vibrio cholerae* (*V. cholerae*), specifically serogroups O1 and O139, survives in aquatic environments and is present in the excreta (stools and vomit) of infected individuals. Infection is acquired by ingesting a sufficient bacterial load from the environment (indirect transmission), or contact with infectious excreta (direct transmission). *V. cholerae* abundance in aquatic reservoirs varies through interactions with biotic and abiotic factors. Elements of aquatic flora and fauna are associated with *V. cholerae* abundance [5]. Concomitantly, environmental parameters including water temperature and salinity also influence the *V. cholerae* life cycle in its aquatic reservoir [6,7]. Viable *V. cholerae* can persist in the environment in suboptimal conditions for over 15 months in a non-culturable state [5], from which it can revert to a culturable state in favorable conditions. Inappropriate waste management can introduce *V. cholerae* in natural or manmade water reservoirs [8,9] and trigger outbreaks through consumption of contaminated water. An outbreak can then be fueled by both direct and indirect transmission as the increased prevalence of the infection can result in contamination of additional water reservoirs. The dominant transmission routes can be hard to disentangle but their identification is critical to control cholera.

The Global Task Force on Cholera Control (GTFCC) has set a road map to eliminate cholera in 20 endemic countries by 2030 [10], defining SSA as an important target. Generally, diseases or pathogens are considered endemic in an area when they display persistent local transmission for an extended period of time. For cholera, the World Health Organization (WHO) defines an area as endemic when local transmission caused confirmed cases in the previous three years [10]. This definition encompasses a wide variety of transmission patterns, which could cause the same intervention to have different impacts in different endemic areas. In non-endemic areas, the environmental contribution to cholera transmission is often low, but in endemic areas the relative contribution of direct and indirect transmission routes is

often unknown. The benefits expected from cholera interventions, as traditionally implemented in outbreak response, become less clear in endemic settings because they do not necessarily target the dominant transmission route.

Cholera transmission can be prevented by improving water and sanitation infrastructures and with vaccination. Water, sanitation, and hygiene (WASH) improvements have historically been the primary prevention tool. WASH improvements are resource- and time-intensive to implement [11]. They are extremely effective; waste management and water infrastructures have largely prevented cholera transmission in high income countries [12]. Large scale WASH improvements are necessary to control cholera [13], however resource scarcities limit such improvements in the countries carrying most of the global burden: SSA nations have some of the poorest access to clean water and improved toilets in the world [14]. In comparison, implementing a vaccination campaign is fast and can reduce cholera transmission quickly. The empirical results of the reactive use of oral cholera vaccines (OCV) in 2012 in Guinea and theoretical results from modeling studies demonstrated the utility of vaccination as a tool to control cholera [15,16]. A quick vaccine rollout leads to a rapid increase in population immunity that can mitigate cholera transmission, but it is a short term solution because the acquired protection declines after about 2–3 years [17]. The increasing stockpile of OCV allowed for more frequent use of vaccines in outbreak response and its novel use in endemic areas [18,19].

Both OCV and WASH improvements are important components of the multisectoral interventions required to control cholera in areas with high burden [10,19]. While the benefit of OCV is straightforward in epidemic settings [20,21], it might be narrow in an endemic setting. The impact of OCV on transmission correlates with the increase in population immunity but immunity may always be high if cholera exposure is frequent and widespread, which can be the case in endemic settings. Quantifying the impact of interventions using OCV in endemic settings could provide valuable information to inform control strategies and achieve the ambitious goals set by the GTFCC.

The Democratic Republic of Congo (DRC) has consistently carried one of the highest cholera burdens in the African Great Lakes region [2]. Cholera is endemic in the Congolese city of Kalemie, in Tanganyika Province, which lies on the shore of Lake Tanganyika (Fig 1A and 1B). The area displays annual peaks of cholera cases, typically during rainy seasons (Fig 1C), and reports suspected cholera cases all year. Lake Tanganyika could act as an environmental reservoir providing frequent exposure. In parallel, the local population is highly mobile with 24.7% of the residents of Tanganyika Province reporting travelling at least once in the previous 12 months for a duration of at least 1 month [22]. The strong fishing activity, with fishermen moving seasonally and experiencing exposure to the lake and low sanitation conditions, may be a potential source of reintroduction [23]. Such mobility could also promote cholera persistence through metapopulation dynamics.

The city of Kalemie received a cholera intervention in 2013–2016 that included both an OCV campaign and limited WASH improvements. The health system in DRC is organized around nested geographical units: Provinces, health zones (HZ), and health areas (HA). Public health interventions are often organized and implemented at least at HZ level. The city of Kalemie spreads across two HZ, Kalemie and Nyemba (Fig 1B). The vaccination campaign targeted HA that were in Kalemie city, where attack rates had historically been the highest as of November 2013. The vaccination campaign originally targeted about 120,000 people in four HA with two doses of Shanchol, but was interrupted after three days due to security issues. It resumed in July 2014 and the expiration of vaccine doses led to reducing the target population to about 52,000 people in two HA. Ultimately, 81.2% of the target population received at least one dose [24]. The WASH component of the intervention focused on improving access to clean water. Although it was not acting on every dimension of WASH, we simply refer to it as "WASH

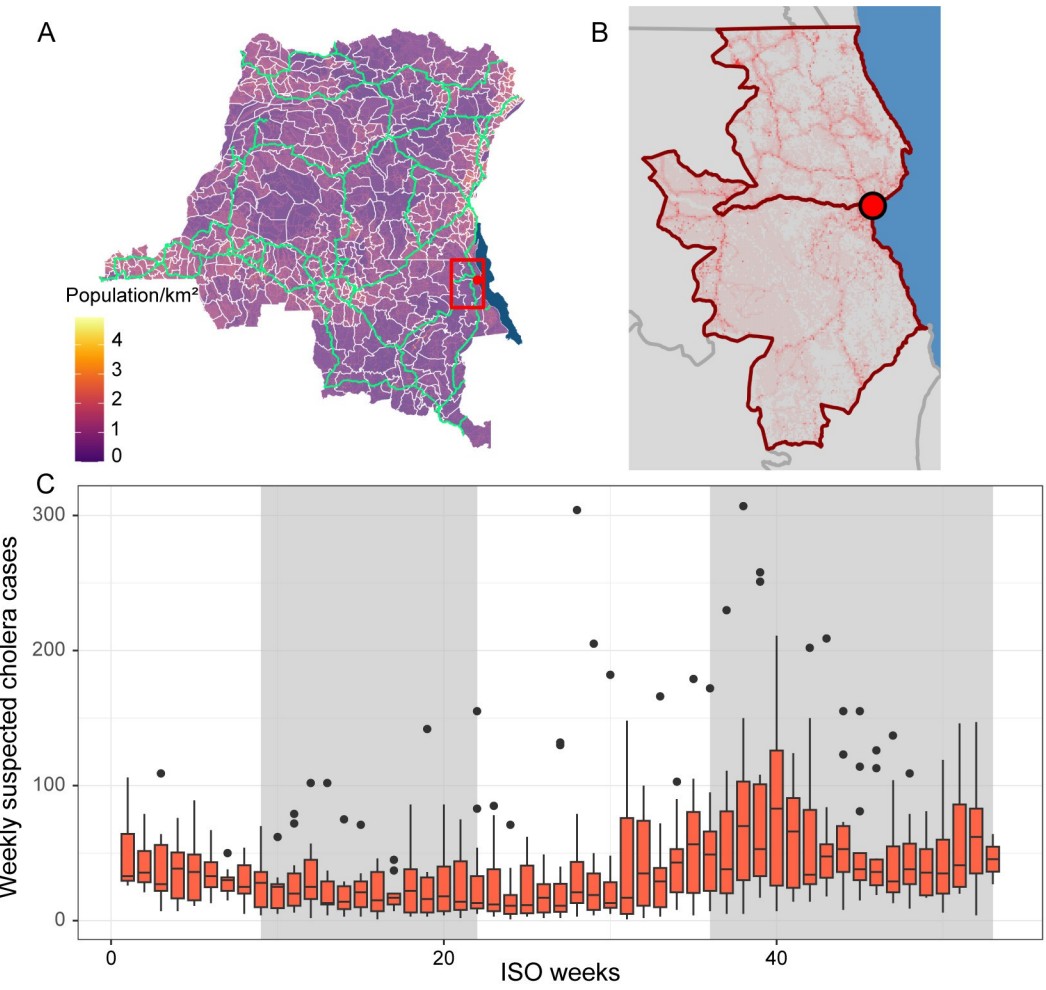

**Fig 1. Overview of the location and the seasonality of cholera cases in the study area.** (A) Map of the DRC with population density (log transformed), boundaries of health zones in white, major roads in light green, a red box around the health zones of Kalemie city, Lake Tanganyika in blue. (B) Detail of red box from A, red circle on Kalemie city, Lake Tanganyika in blue. Low population density in grey, high in red. Health zones of Kalemie and Nyemba outlined in dark red. (C) Weekly number of reported suspected cholera cases (based on the International Organization for Standardization (ISO) system) in the health zones of Kalemie and Nyemba from 2002 to 2014 [25], typical rainy season weeks shaded in grey. Direct links to the map layers are: https://www.gadm.org/download_country.html, https://www.globio.info/global-patterns-of-current-and-future-road-infrastructure, https://landscan.ornl.gov, https://wbwaterdata.org/dataset/africa-water-bodies-2015.

intervention" below. Doctors Without Borders (*Médecins Sans Frontières*, MSF) extended access to tap water in the northern part of the city by laying pipes, building water reservoirs, distributing water filters, and setting up public drinking fountains in collaboration with *Solidarites International*. In addition, sand filters were installed on paths where people draw water from the lake, and chlorination activities were performed during outbreaks. The WASH intervention incurred delays in the aftermath of the security issue that delayed the OCV intervention. Its first milestone, extending access to tap water was achieved in October 2014 and the remaining components were completed incrementally until early 2016.

We fit a group of deterministic compartmental models that included interhuman cholera transmission with and without environmental contribution and seasonal migration. We used the model with the best fit to assess the short-term impact of this multi-pronged intervention

in the city of Kalemie while considering the potential influence of environmental drivers and their contributions to local transmission.

## Methods

### Ethics statement

The Ethical Review Board of the University of Lubumbashi approved the study protocol to assess the impact of the vaccination campaign (study protocol ethical number: UNILU/CEM/028/2013) and its extension (study protocol ethical number: UNILU/CEM/050/2015). Individuals provided oral informed consent to be part of the vaccine coverage survey. If the participants were minor, oral consent was obtained from the parent/guardian. Pennsylvania State University's Institutional Review Board determined the post-intervention handling and analyses of these anonymized data was not Human Research (STUDY00015621).

### Method overview

We fit a group of Susceptible-Infected-Recovered-Susceptible models with a compartment, B, for the bacterial population in the environmental reservoir (SIRB), Lake Tanganyika [26]. We explored the influence of seasonal migration on cholera transmission by fitting models with different structures: with both susceptible and infected (in bold in Eqs 1, 2, 5, and 6 below), or only susceptible individuals migrating, or no migration.

We fit the SIRB models to the reported suspected cholera cases presenting at the only cholera treatment center in the city of Kalemie from November 2013 to February 2016. For this period of time only, detailed surveillance data were gathered in an electronic register with support from MSF as part of a study to assess the impact of the intervention. Only residents of the city of Kalemie were included in the analysis.

The structure of the full model is as follows:

$$\frac{dS}{dt} = \delta R - \beta_h \frac{SI}{N} - \beta_e \frac{B}{\kappa + B}(1 + \lambda_e f(rain_t))S - \eta S + \boldsymbol{f_s(rad_t)} \tag{1}$$

$$\frac{dI}{dt} = \beta_h \frac{SI}{N} + \beta_e \frac{B}{\kappa + B}(1 + \lambda_e f(rain_t))S - \gamma I + \boldsymbol{f_I(rad_t)} \tag{2}$$

$$\frac{dR}{dt} = \gamma I - \delta R + \eta S \tag{3}$$

$$\frac{dB}{dt} = \mu(1 + \lambda_e f(rain_t))I - B(\varepsilon - \varphi_t) \tag{4}$$

with

$$\boldsymbol{f_s(rad_t)} = \boldsymbol{\alpha_1 rad_t} \tag{5}$$

$$\boldsymbol{f_I(rad_t)} = \boldsymbol{\alpha_I rad_t} \tag{6}$$

$$ratio_{S/I} = \frac{\alpha_1}{\alpha_I} \tag{7}$$

$$\varphi_t = e^{\alpha_2 + \alpha_3 sst_t + \alpha_4 chlor_t} \tag{8}$$

$$\eta = (\sigma_1 VC_{1t} + \sigma_2 VC_{2t})\ddot{\Upsilon} \tag{9}$$

$$f(rain_t) = \frac{rain_t}{max(rain_t)} \tag{10}$$

The interpretation of all the parameters of the full model is presented in Table 1 and is described in more details below. Susceptible individuals become infected through exposure to the environmental reservoir, $\beta_e$, or through interhuman transmission, $\beta_h$. The WASH intervention decreased the environmental exposure rate $\beta_e$ to $\beta_e–\beta_{WASH}$ by the end of the study period. $\beta_e$ was assumed to decrease linearly from $\beta_e$ to $\beta_e–\beta_{WASH}$ from the time the first component of the WASH improvements was completed (ISO week 40 in 2014). The model did not allow the environmental contamination to vary because the intervention did not target waste management. The infection probability from an exposure to the environment followed a dose-effect relationship, with the half saturation constant $\kappa$. Infected individuals transitioned to the recovered compartment at rate $\gamma$. Susceptible individuals could gain immunity through vaccination, $\eta$, 1 week after receiving the vaccine [15]. This was included through a step function of the number of people who received 1 or 2 doses ($VC_{1t}$ and $VC_{2t}$) of Shanchol. We estimated the number of vaccinated individuals from vaccine coverage estimates from a survey performed by MSF [24] and the associated population size estimates (see S1 Text). We considered a range of values for vaccine effectiveness for one and two dose regimens ($\sigma_1$ and $\sigma_2$), including

**Table 1. Interpretation on the model parameters.**

| Parameter | Description |
|---|---|
| $\beta_h$ | Infection rate related to interhuman transmission |
| $\beta_e$ | Infection rate related to environmental exposure |
| $\beta_{WASH}$ | Reduction in infection rate related to environmental exposure due to the WASH intervention |
| $\delta$ | Recovery rate |
| $\kappa$ | Half saturation constant |
| $\alpha_I$ | Intensity of the net flux of infected individuals as function of the variation of anthropogenic nighttime radiance |
| $\alpha_1$ | Intensity of the net flux of susceptible individuals as function of the variation of anthropogenic nighttime radiance |
| $\alpha_2$ | Baseline bacterial growth in the environment (exponential scale) |
| $\alpha_3$ | Strength of the association between lake surface temperature variation and bacterial growth variation (exponential scale) |
| $\alpha_4$ | Strength of the association between chlorophyll-a variation and bacterial growth variation (exponential scale) |
| $\lambda_e$ | Strength of the amplification of the environmental exposure due to precipitation |
| $\lambda_c$ | Strength of the amplification of the environmental contamination due to precipitation |
| $\delta$ | Immunity waning rate |
| $\mu$ | Environmental contamination rate |
| $\varepsilon$ | Bacterial decay rate |
| $S_0$ | Susceptible individuals at time 0 |
| $I_0$ | Infectious individuals at time 0 |
| $B_0$ | Number of cholera bacteria at time 0 |
| $r$ | Reporting proportion |
| $\psi$ | Overdispersion parameter |

estimates from studies done in the aftermath of reactive vaccination campaigns performed in Zambia and Guinea [15,27] (see Table B in S1 Text). Our models assumed an all-or-nothing effect of vaccination, implying optimistic estimates of its impact, but we also fit an alternative model structure with a leaky vaccine as sensitivity analysis (see Table H, and Fig O and P in S1 Text). Considering the wide age range of the target population (everyone older than 1 year), we assumed that the proportions of susceptible, infected, and recovered among the vaccinated individuals were the same as the general population when the doses were distributed. Immunity waned at rate $\delta$, returning immune individuals to the susceptible compartment. We did not include booster effects on immune individuals receiving vaccine. Booster effects are unlikely to be detected in the study period of 118 weeks (most doses were distributed on the 32$^{nd}$ and 35$^{th}$ week), because the study period is shorter than the average period of immunity, whether acquired through infection or vaccination [28,29]. We also assumed that vaccination had no impact on those who were infected at the time of vaccination. We added a penalty term ($\ddot{\Upsilon}$) to account for the spatially targeted nature of the vaccination campaign, which focused on HA in the city of Kalemie with historically high attack rates, where residents had experienced more cholera exposure, further decreasing the proportion of susceptibles. We considered a range of possible values for $\ddot{\Upsilon}$ (between 0.7 and 1) (see S1 Text).

Population size was allowed to vary through seasonal migration ($f_S(rad_t)$ and $f_I(rad_t)$), which can influence local cholera transmission through regular reintroductions from areas with ongoing transmission. We included migration by quantifying the seasonal variation of contemporaneous anthropogenic nighttime radiance, extracted from Visible Infrared Imaging Radiometer Suite (VIIRS) data [30] (see S1 Text). We assumed that the net migration flow varied linearly with the first derivative of the nighttime radiance data in the area ($rad_t$) [31]. We first fit a generalized additive model with a cyclical spline to the radiance data and then extracted its first derivative (see S1 Text). We did not consider the mobility of immune individuals, because they do not actively contribute to transmission. We considered a range of values for the ratio of susceptible and infectious individuals among the mobile population ($ratio_{S/I}$) (between 10 and 100) (see S1 Text). We explored alternative model structures allowing only susceptible individuals to be mobile ($f_I(rad_t) = 0$) or removing seasonal mobility ($f_S(rad_t) = 0$ and $f_I(rad_t) = 0$) (see S1 Text).

We considered the influence of water temperature, with lake surface temperature ($SST_t$), and phytoplankton, with chlorophyll-a ($chlor_t$), as environmental drivers on aquatic bacterial growth [5]. We extracted these values from Moderate Resolution Imaging Spectroradiometer data [32] (see S1 Text). Precipitation ($rain_t$) could also increase exposure to environmental reservoir and its contamination with infectious human excreta by respectively contaminating drinking water sources [33] ($\lambda_e f(rain_t)$) and flooding defecation sites ($\lambda_c f(rain_t)$). We extracted precipitation estimates from meteorological forcing data [34].

The bacterial population in the environment increased with contamination of the lake from the excreta of infected individuals ($\mu$), and a time dependent bacterial growth rate ($\varphi_t$) that varied with $SST_t$ and $chlor_t$. Conversely, it decreased through constant bacterial decay ($\varepsilon$).

The models did not include births, deaths, or the age structure of the host population because of the short study period of 118 weeks. Based on case management and a CFR of 0.3% during this 118 week-period (5 deaths reported among the 1634 resident suspected cholera cases), we did not include cholera specific mortality.

We used a negative binomial process to link the predicted number of weekly incident cases ($C_t$) and the weekly reported suspected cases ($A_t$) : $A_t \sim NegBinom(C_t r, C_t \psi)$, with r, a combination of reporting rate and the portion of true cases captured by the suspected case definition (see S1 Text), assumed constant, and $C_t \psi$, an overdispersion parameter scaling with the predicted number of new cases. The negative binomial distribution can handle overdispersion

and its scaling overdispersion parameter allows variance estimates to better scale with fast and large variations of the incidence.

Using different assumptions regarding $ratio_{S/I}$, $\sigma_1$, $\sigma_2$, and $\ddot{\Upsilon}$, we fit a group of 96 models: 64 variations of the full model, 16 variations of the model with only susceptible individuals migrating, and 16 variations of the model without seasonal migration (see Table C and Fig D in S1 Text). We assessed model fit with the widely applicable information criteria (WAIC) [35] and selected the best performing model presented here with the lowest WAIC or with fewer parameters for similar WAIC. We also performed a sensitivity analysis of the best performing model by removing the possibility for bacterial growth ($\varphi_t = 0$) or the environmental compartment and indirect transmission ($\beta_e = 0$) (see Fig E in S1 Text).

We estimated the parameters $\beta_h$, $\beta_e$, $\beta_{WASH}$, $\alpha_1$, $\alpha_2$, $\alpha_3$, $\alpha_4$, $\lambda_e$, $\lambda_c$, $\delta$, $\mu$, $\varepsilon$, r, and $\psi$, and the initial conditions $S_0$, $I_0$, $B_0$ through Markov chain Monte Carlo sampling using the Metropolis-Hastings algorithm. All the estimates presented are the mean values over the posterior distribution and their 95% credible interval (95% CrI) using the highest density interval.

We assessed the short-term impact of each arm of the intervention separately and both arms together by estimating the number of additional cases in their absence. We fixed $\eta$ to 0 while keeping $\beta_{WASH}$ unchanged, simulating WASH improvements without vaccination, did not allow $\beta_e$ to decrease ($\beta_{WASH} = 0$) while keeping $\eta$ unchanged, simulating vaccination without WASH improvements, and then fixed both $\eta$ and $\beta_{WASH}$ to 0, simulating no vaccination and no WASH improvements. We sampled 10,000 sets of parameters from the posterior distribution and calculated the number of additional cases in each of the alternative scenarios compared to the intervention as it happened.

We explored alternative vaccination strategies by varying the timing and the size of the target population, between 50,000 to 200,000 (19.0–76.1% of the population of the city of Kalemie), assuming one campaign during the 118-week period with a two-dose regimen (without WASH). The maximum target population size considered is within the MSF vaccination capacity observed in other settings [27]. We estimated the number of cases avoided for each scenario by calculating the reduction in cholera cases compared to no intervention for each of 10,000 set of parameters sampled from the posterior distribution. We considered 84 combinations of alternative timing and target population size. We sampled 500 sets of parameters for each combination, computational intensity prohibited more.

We investigated the relative contributions of environmental exposure and contamination to transmission assuming no intervention by simulating scenarios with no environmental exposure ($\beta_e = 0$), or no environmental contamination ($\mu = 0$) and calculating the number of additional cases compared to having them both ($\beta_e$, and $\mu$ unchanged) for each of 10,000 sets of parameters sampled from the posterior distribution.

## Results

Models with no seasonal migration had a comparable fit to the ones with only susceptible individuals migrating or seasonal migration of both susceptible and infected individuals (see Table C and Fig D in S1 Text). This suggested that mobility had minimal influence on the observed cholera dynamics. We selected the model with the lowest WAIC among the ones without seasonal migration, which had fewer parameters. It reproduced the reported weekly cholera cases well, with 98.3% (116/118) of the observed data in the model prediction's envelope of the 95% CrI of weekly reported suspected cases (Fig 2A). The model suggested high local immunity, fluctuating between 88.8% and 99.9% (Fig 2B). This high immunity would be the likely consequence of annual outbreaks and persistent environmental exposure, which we explain further below. Based on our model, the targeted vaccinations occurred when

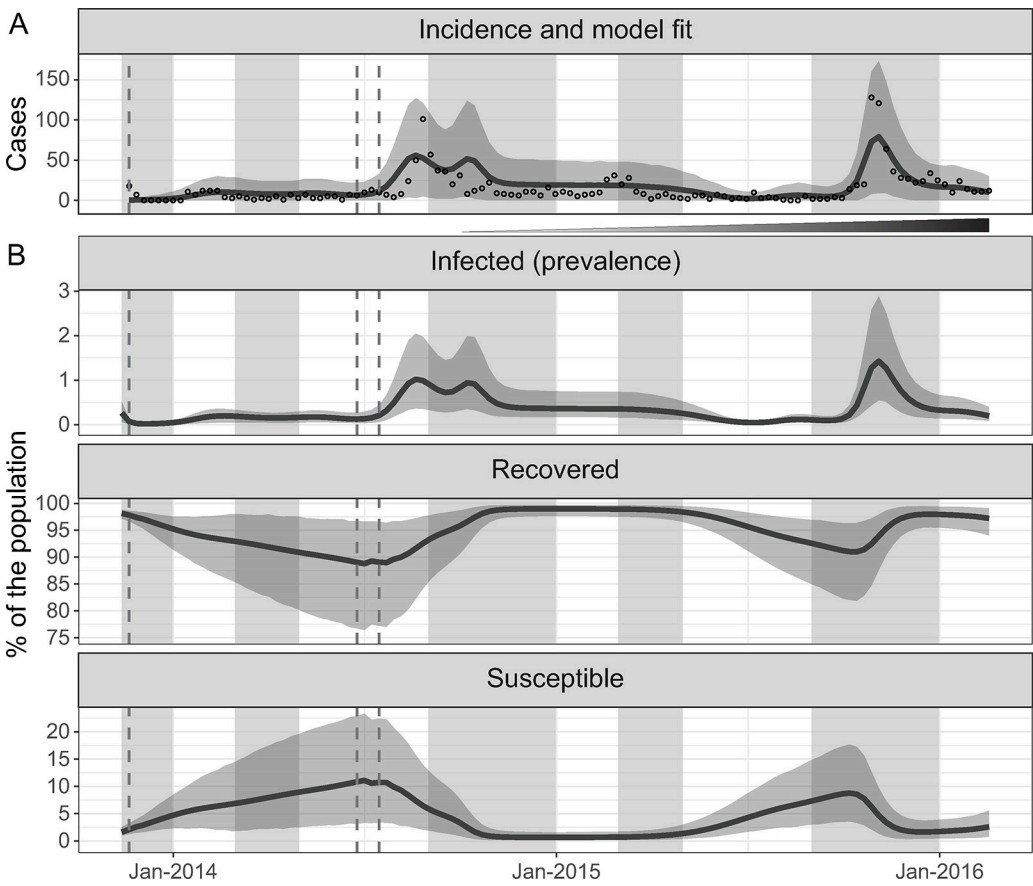

**Fig 2. Incident cases, model fit, and variation of the percentage of infected, recovered, and susceptible over time.** (A) Weekly reported suspected cholera cases residing in the city of Kalemie (empty circles) from November 2013 to February 2016 and mean model prediction of the reported weekly cholera cases (dark line) and its 95% credible interval (grey envelope) (B) Mean model estimates of the percent age of the population infected (prevalence), recovered, and susceptible (dark lines) and their 95% credible interval (grey envelopes) from November 2013 to February 2016. Typical rainy seasons are shaded in grey, the timing of the distribution of vaccine doses in vertical dashed grey lines, and the incremental implementation of the improvements in water and sanitation is indicated by the widening and darkening triangle between A and B.

population immunity was high: 97.8% (95% CrI: 96.7–98.6) in November 2013, 89. 0% (95% CrI: 76.7–96.7) in July 2014, and 89.1% (95% CrI: 77.2–96.6) during the catch-up in August 2014.

Both the scenarios omitting vaccination (WASH only, and no WASH and no vaccination) visibly lacked a reduction in the susceptible proportion of the population in July 2014 (Fig 3A, bottom panel). Over this 118 week period, we estimated: 3,702 (mean: 3,702.3, 95% CrI: 1,302.5–7,542.0) additional cases, a 2.56% increase (mean: 2.56%, 95% CrI: 1.79%-3.30%), when removing vaccination alone (scenario with WASH only), 1,585 (mean: 1,585.5, 95% CrI: 1,321.9–5,108.8) cases avoided, 1.03% (mean: 1.03%, 95% CrI: 0.01%-2.85%), by WASH alone (scenario with vaccination only), and 5,259 (mean: 5,258.6, 95% CrI: 1,576.6–11,337.8) cases avoided, 3.57% (mean: 3.57%, 95% CrI: 2.02%-5.72%), by implementing both vaccination and WASH (scenario with no vaccination and no WASH improvements) (Fig 3B).

Our model suggested that vaccination campaigns with small target population sizes would have a limited impact in populations with high immunity (Fig 3C). However, the timing of a pulse of vaccination could substantially influence the impact of vaccination campaigns.

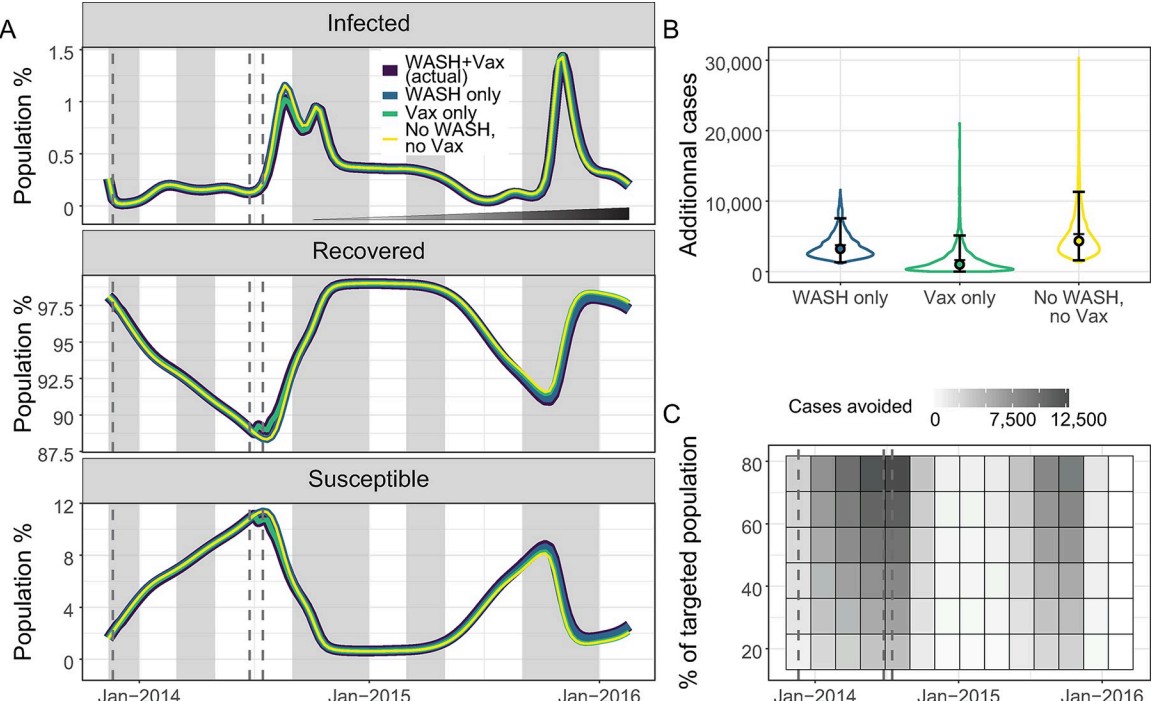

**Fig 3. Estimated impact of the components of the intervention and impact of alternative vaccination strategies.** (A) Mean model predictions of the percentage of infected, recovered, and susceptible in the population considering: the intervention as it happened of WASH and vaccination (dark blue), WASH only (blue), vaccination only (green), and no intervention of either WASH or vaccination (yellow) from November 2013 to February 2016. The incremental implementation of the improvements in water and sanitation is indicated by the widening and darkening triangle in the top panel. Typical rainy seasons are shaded in grey and the timing of the distribution of vaccine doses is shown in dashed grey lines. (B) Violin plots of numbers of additional cholera cases at the end of the study period with WASH only (blue), vaccination only (green), or no intervention (no WASH and no vaccination) (yellow) compared to the intervention as it happened for WASH and vaccination. The error bars, the filled circles, and the horizontal bars indicate the 95% credible interval, the medians, and the means respectively. (C) Heatmap of the mean number of cases avoided by changing the timing and the coverage of a vaccination campaign compared to a scenario without intervention. The timing of the distribution of vaccine doses for the intervention as it happened is indicated by dashed grey lines.

Specifically, timing the vaccination to occur at the lowest point of population immunity and before an outbreak began increased its impact. The best performing vaccination scenario (darkest cell of the heatmap in Fig 3C) avoided 12,777 cases (mean: 12,776.7, 95%CrI: 4,681.0.7–26,019.5) over 118 weeks for 200,000 vaccinated people. However, the high level of local immunity would result in vaccinating a large proportion of immune individuals, reducing the impact of the vaccination.

We estimated that removing environmental exposure or contamination would have a critical impact on cholera dynamics. These strategies avoided 142,518 cases (mean: 142,518.3, 95% CrI: 36,670.0–303,068.3) and 134,373 cases (mean:134,372.8, 95% CrI: 30,921.5–266,103.8), respectively. In each of these scenarios local cholera transmission was virtually interrupted (Fig 4A and 4B). Environmental contamination appeared necessary to maintain a bacterial load sufficient to support environmentally-driven transmission because the fluctuation of *V. cholerae* population averaged towards net decay (Fig 4D, right).

The high immunity inferred by the model was maintained through annual flare-ups and constant environmental exposure. The environmental component of the force of infection ($\Phi_e = \beta_e \frac{B}{\kappa + B}(1 + \lambda_e f(rain_t))$) was consistently greater than the interhuman transmission component ($\Phi_h = \frac{\beta_h I}{N}$) despite $\beta_h$ being greater than $\beta_e$ (Fig 4C). $\Phi_h$ remained low because epidemic

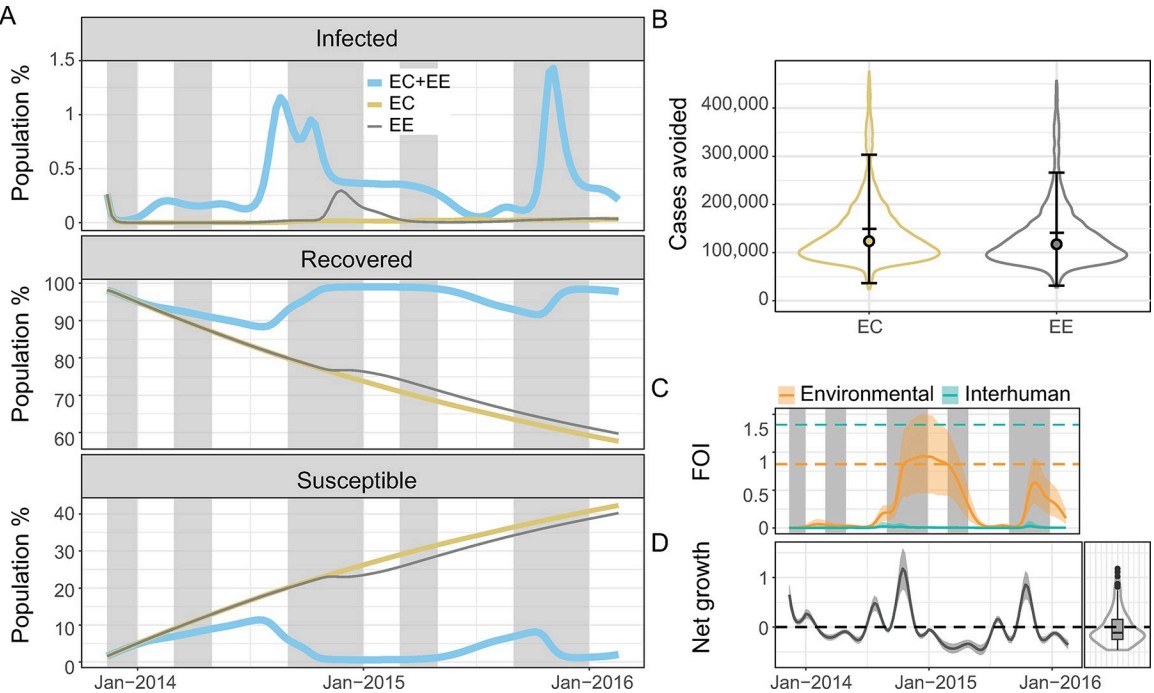

**Fig 4. Contributions of the environmental reservoir in cholera transmission.** (A) Mean model predictions of the percentage of infected, recovered, and susceptible individuals in the population with environmental contamination and exposure (EC+EE) (light blue), environmental contamination only (no environmental exposure) (EC) (beige), and environmental exposure only (no environmental contamination) (EE) (grey) from November 2013 to February 2016. Typical rainy seasons are shaded in grey. (B) Violin plots of numbers of cholera cases avoided by the end of the study period with only EC (beige), or only EE (grey) compared to a scenario with EC and EE. The error bars, the filled circles, and the horizontal bars indicate the 95% credible interval, the medians, and the means respectively. All the scenarios considered in A and B assume that no intervention occurred. (C) Mean prediction of the variation of the environmental (light orange line) and interhuman (light green line) components of the force of infection and their 95% credible interval (light orange and light green envelopes) from November 2013 to February 2016. The light orange and light green dashed lines indicate the mean values of the environmental exposure rate ($\beta_e$) and interhuman transmission rate ($\beta_h$), respectively. (D) Left: Mean prediction of the variation of the environmental net bacterial growth ($\varphi_t - \varepsilon$) (dark line) and its 95% credible interval (grey envelope) from November 2013 to February 2016. Right: Violin plot and boxplot of the distribution of the mean prediction of the net bacterial growth rate from November 2013 to February 2016. The dashed black line indicates 0: values below 0 show net decay and values above 0 show net growth.

flare-ups did not lead to a high prevalence of infection, the way they would in a mostly susceptible population. Conversely, $\Phi_e$ strongly increased with pulses of net bacterial growth due to environmental drivers, despite an overall trend favoring net decay (Fig 4D left and right).

## Discussion

Based on our model, the impact of the intervention performed in Kalemie was modest when measured by cases avoided, preventing an estimated 5,259 cases (mean: 5,258.6, 95% CrI: 1,576.6–11,337.8) for both intervention arms combined. The reduction of the target population size following the interruption of planned vaccination activities, the limited scale and the incremental implementation of the WASH improvements, and the high level of population immunity likely all contributed to mitigating the impact of the intervention.

Benefitting from vaccination in endemic cholera settings, as defined by WHO, requires an understanding of dominant local transmission routes. Our model suggests that the impact of vaccination is small in settings where an environmental reservoir provides constant exposure and maintains high immunity, despite an optimistic assumption of an all-or-nothing vaccine. However, endemicity is more nuanced than the current WHO definition suggests and OCV

could still play an important role in some endemic settings. The inability to identify and target the susceptible individuals would lead to vaccinating a majority of immune individuals in this situation. Achieving very high vaccine coverage would immunize a greater number of susceptible individuals, but at the cost of giving many additional doses to immune individuals. This cost could be reduced by targeting the age group most represented among susceptibles or by guiding vaccination with serosurveys. The age profile of the suspected cholera cases residing in Kalemie (median age of 15 years, and interquartile range (IQR) of 3–34 years) during this period would support restricting the maximum age of the target population to increase the impact of the vaccination campaign. However, defining a meaningful age group target would require high resolution historical epidemiological data, and those would only provide information on symptomatic cases, a portion of infected cases. Similarly, guiding vaccination efforts with serosurveys to target susceptibles would incur a substantial additional cost in addition to the difficulty of applying a binary interpretation to serosurvey results.

Our estimates of average immunity duration ($1/\delta$ = 3.7 years, 95% CrI: 1.8–8.0 years) and the cumulative incidence converted into an average yearly incidence rate (24.1%, 95% CrI: 11.7–47.5) are consistent with current knowledge of post-infection immunity and other incidence rate estimates in another well studied cholera endemic area, Bangladesh. Challenge studies have demonstrated that immunity lasts at least 3 years after natural infection [36]. National incidence rate in late 2015 in Bangladesh was estimated at 17.3% based on a representative survey and analyses of vibriocidal titres [37].

Our model suggests that a well-timed large-scale vaccination could improve the impact of vaccination in the city of Kalemie, potentially avoiding an average of 12,777 cases (95%CrI: 4,681.0–26,019.5) for 200,000 vaccinated individuals. However, this requires implementing a large vaccination campaign with precise timing. It would be logistically challenging and costly to implement vaccination campaigns of this scale with very precise timing, dictated by the need to vaccinate when immunity is at its lowest and before environmental drivers trigger a pulse of force of infection. This approach would still achieve only short term and small-scale benefits. On the other hand, our findings suggest that WASH improvements on a scale large enough to prevent environmental exposure and contamination for the whole population could have a dramatic long-term impact. Although we estimated that the WASH improvements in Kalemie prevented a modest number of cases, this is likely partially due to the short period of time considered to assess the impact of this part of the intervention. The main components of this WASH intervention consisted of extending the pipe network and building a water reservoir, and they were completed incrementally during the 118-week period. While extending access to the pipe network is an important step, it does not guarantee reliable and consistent access to chlorinated tap water [38]. The magnitude of the improvements required to ensure both access to safe water and efficient waste management, not only in Kalemie but throughout the cholera-affected nation of DRC, appears immense but necessary to control cholera. Implementing WASH improvements should be considered a priority not only to control cholera, but also to prevent the transmission of other water-borne and fecal-oral pathogens that contribute to the disease burden in DRC [39]. This approach will also help achieve the 6[th] goal of the Sustainable Development Goals [40], to ensure availability and sustainable management of water and sanitation for all, in a country where WASH improvements are critically needed [14].

Kalemie is not unique regarding a potentially strong environmental driver of cholera transmission. Substantial environmental contributions for cholera cases have been reported in Haiti and Zimbabwe, areas where the basic reproduction number was estimated to rely mostly on its environmental component [16]. Environmental drivers are also important drivers in other endemic settings like Bangladesh and India, although they act differently: flooding in the

early and late phase of the monsoon is strongly associated with higher cholera incidence [41], while the peak of the monsoon is associated with a cholera lull due the "dilution" of *V. cholerae* in its reservoir [42]. Although mobility does not appear to be necessary for local cholera persistence in the city of Kalemie, movement could make Kalemie a source of cholera that can seed outbreaks in surrounding areas that lack an environmental source and where exposure is less frequent. The older ages of the suspected cholera cases residing outside of Kalemie (median age of 24.5 years, and IQR: 5.75–39.25 years) are consistent with lower exposure rates and source-sink dynamics.

Our estimates supporting a major role of environmentally driven transmission in Kalemie's local cholera dynamics appear plausible. Sensitivity analysis showed that removing the environmental component or bacterial growth of the model significantly decreased its ability to fit the observed data (see Fig E in S1 Text). Confirming our estimates regarding population immunity and the dominant source of bacterial infection would require further research including serosurveys and substantial microbiological monitoring of the lake water in the area. Serological data would allow us to support or disprove our findings, however such data are currently lacking in DRC. Evidence of environmental presence of toxigenic *V. cholerae* is scarce in the area. Extensive water sampling in Lake Tanganyika from October 22nd to 26th 2018 did not detect toxigenic *V. cholera* [43]. However, it was detected in ten environmental samples, in fish and water, also collected from Lake Tanganyika from October 2018 to March 2019 and there is some evidence of increased positive samples during rainy seasons in other environmental sampling studies [44–46].

We estimated that the natural variation of the *V. cholerae* population in the lake leans in favor of net decay. Previous modeling studies assumed bacterial growth rates to consistently be in favor of net decay, whether they varied over time or not [47,48]. More recent studies considered the possibility for complex bacterial growth patterns but were entirely theoretical [49]. Our model allowed environmental bacterial abundance to vary based on environmental inputs, leading to temporary switches to net bacterial growth. These were important in creating pulses of high environmentally-driven force of infection. Improving the quality of consumed water (reducing environmental exposure) had a large impact in our simulations and removing environmental contamination had an impact almost as large. The overall trend toward net bacterial decay in our model highlights that regularly replenishing local bacterial population through environmental contamination is potentially a critical component of local persistence. This emphasizes the potential compounded benefits of comprehensive improvements to sanitary infrastructures and access to clean water.

We did not consider cholera-induced mortality because of the low number of cholera-induced deaths in this population and the local experience in managing cholera infections. However, there is evidence that a substantial portion of cholera mortality occurs in the community [50], so we cannot rule out that some cholera-induced mortality is not captured in the reported data. The lack of data on mortality in the community prevented us from estimating the number of deaths avoided by the intervention. Our model made simplifying assumptions regarding immunity: we did not account for the various levels of protection acquired after an infection with or without symptoms and did not consider a booster effect of the vaccination on already immune individuals. While immunity is likely shorter after an infection with no or mild symptoms, very little is known about these dynamics because they are difficult to measure [36]. Our model likely presents a transmission pattern averaged across several infectious states that contribute variably to the force of infection; we did not attempt to explicitly separate them due to the absence of data to guide the necessary assumptions. Not considering the booster effect of vaccination on already immune individuals could have led us to slightly underestimate the duration of immunity but this is unlikely to impact our estimates considering the

short study period (118 weeks) compared to our estimated average immunity period (3.7 years). We also assumed that immunity wanes at similar rates for susceptible individuals who were successfully vaccinated and following natural infection, but vaccine-induced immunity likely wanes faster [51]. This would have little impact on our estimates considering the small proportion of susceptible individuals in the population when doses were distributed in our model as well as short study period [52]. However, such additions in the model structure would be necessary for a longer time series as the impact of these simplifications would increase.

We included the potential impact of the WASH intervention in a simplistic way, assuming a linear variation of the environmental transmission rate. This approach required the fewest additional assumptions; a more refined and detailed method would likely improve the validity of our estimates but the data required to do this do not exist. To estimate environmental drivers, we used measurements of chlorophyll-a and surface water temperature in the lake in addition to the influence of rain. The interactions between *V. cholerae* and other elements of its aquatic reservoir are only vaguely understood [5,53]. We modeled one common environmental reservoir, as is customary in SIRB models, which implies that our estimates could hide spatially heterogeneous environmental exposures. Ultimately, we cannot assess how accurately we captured the main fluctuations of the environmental bacterial population in the absence of thorough environmental sampling in the area. However, we considered only environmental drivers that have been associated with *V. cholerae* environmental abundance or exposure to the environmental reservoir. Phytoplankton growth, indirectly measured through chlorophyll-a, has been associated with cholera outbreaks in several studies, and specifically cyanobacteria are a credible reservoir for *V. cholerae* [5,54]. Water temperature influences phytoplankton growth [5], and the consequence of rainfall on environmental exposure and environmental contamination to/from *V. cholerae* is credible in this setting along a lake with low access to water and sanitation infrastructures [33]. We explicitly included a direct proxy of human presence, anthropogenic nighttime radiance, in the models considering seasonal mobility. Nighttime radiance is a reliable indicator of human presence and has been used to infer population mobility in both high and low-income countries [31,55,56]. We are confident that we robustly captured the seasonal migration and the environmental components in our model, though there might be limitations in the spatiotemporal resolution and availability of remote sensing data, particularly for chlorophyll-a.

Impact assessments of cholera interventions are scarce in endemic settings, particularly beyond estimates of vaccine effectiveness and vaccine coverage. Studies like this one are crucial to guide cholera elimination. OCV and WASH improvements are core components of the toolbox to control or eliminate cholera. However, the value of OCV for reactive vaccination in epidemic settings has not been clear in areas with various patterns of endemicities. The assumption that most of the target population is susceptible becomes less accurate as transmission is increasingly environmentally driven. Reducing cholera transmission in endemic areas will require a location-specific understanding of the transmission routes to tailor a strategy; a "one size fits all" approach is unlikely to achieve satisfying results. Geographically-coordinated strategies that target location-specific transmission dynamics might also be necessary to achieve regional cholera control.

## Supporting information

**S1 Text. Additional analyses on some of the assumptions of the model, additional information on the methods and the results, and details on the convergence checks.**
(DOCX)

## Acknowledgments

We would like to acknowledge support and feedback from Nathan Wikle on statistical analyses.

## Author Contributions

**Conceptualization:** Alexandre Blake, Francisco Luquero, Didier Bompangue, Nita Bharti.

**Data curation:** Alexandre Blake, Adam Walder, Francisco Luquero.

**Formal analysis:** Alexandre Blake, Adam Walder, Francisco Luquero.

**Investigation:** Alexandre Blake, Francisco Luquero, Didier Bompangue.

**Methodology:** Alexandre Blake, Adam Walder, Ephraim M. Hanks, Francisco Luquero, Nita Bharti.

**Software:** Alexandre Blake, Adam Walder.

**Visualization:** Alexandre Blake, Nita Bharti.

**Writing – original draft:** Alexandre Blake, Nita Bharti.

**Writing – review & editing:** Alexandre Blake, Adam Walder, Ephraim M. Hanks, Placide Okitayemba Welo, Francisco Luquero, Didier Bompangue, Nita Bharti.

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
