## [Decision Letter · Decision Letter 0]

6 Dec 2024

PNTD-D-24-01300

Impact of a multi-pronged cholera intervention in an endemic setting

Dear Dr. Blake,

Thank you for submitting your manuscript to PLOS Neglected Tropical Diseases. After careful consideration, we feel that it has merit but does not fully meet PLOS Neglected Tropical Diseases's publication criteria as it currently stands. Therefore, we invite you to submit a revised version of the manuscript that addresses the points raised during the review process.

Please submit your revised manuscript within 60 days Feb 04 2025 11:59PM. If you will need more time than this to complete your revisions, please reply to this message or contact the journal office at plosntds@plos.org. Please include the following items when submitting your revised manuscript:

We look forward to receiving your revised manuscript.

Kind regards,

Joseph M. Vinetz

Section Editor

Joseph Vinetz

Section Editor

Shaden Kamhawi

co-Editor-in-Chief

Paul Brindley

co-Editor-in-Chief

**Journal Requirements:**

1) We do not publish any copyright or trademark symbols that usually accompany proprietary names, eg ©,  ®, or TM  (e.g. next to drug or reagent names). Therefore please remove all instances of trademark/copyright symbols throughout the text, including:

- TM on pages: 6, and 9.

2) We have noticed that you have uploaded Supporting Information files, but you have not included a list of legends. Please add a full list of legends for your Supporting Information files after the references list.

**Reviewers' Comments:**

Reviewer's Responses to Questions

**Key Review Criteria Required for Acceptance?**

**Methods**

-Are the objectives of the study clearly articulated with a clear testable hypothesis stated?

-Is the study design appropriate to address the stated objectives?

-Is the population clearly described and appropriate for the hypothesis being tested?

-Is the sample size sufficient to ensure adequate power to address the hypothesis being tested?

-Were correct statistical analysis used to support conclusions?

-Are there concerns about ethical or regulatory requirements being met?

Reviewer #1: The objectives of the study are clearly articulated.

I think that several additional analyses are necessary to the importance of the environmental reservoir and the high level of population immunity estimated in this study. Further details are provided in the general comments below.

**Results**

-Does the analysis presented match the analysis plan?

-Are the results clearly and completely presented?

-Are the figures (Tables, Images) of sufficient quality for clarity?

Reviewer #1: Results are clearly presented.

**Conclusions**

-Are the conclusions supported by the data presented?

-Are the limitations of analysis clearly described?

-Do the authors discuss how these data can be helpful to advance our understanding of the topic under study?

-Is public health relevance addressed?

Reviewer #1: The conclusions could be better supported with some additional analyses as described in my general comments below.

**Editorial and Data Presentation Modifications?**

Reviewer #1: (No Response)

**Summary and General Comments**

Reviewer #1: This study uses mathematical modeling to assess the impact of vaccination and WASH improvement campaigns in a cholera endemic setting in eastern DRC, and also uses the model to assess the potential impact of optimal intervention strategies. The model is also used to assess the relative importance of direct transmission vs. indirect transmission via a contaminated environmental reservoir. The estimated impact of the vaccination campaign was relatively small, in large part because the model estimates that a large fraction of the population was immune prior to the outbreak. My main concern is that there is no serological or other evidence that would support this estimate, and if this estimate is off due to factors not considered in the model (heterogeneity in exposure, risk, etc) then the estimates of both the impact of vaccination and the relative importance of the environmental reservoir might be incorrect.

One possible check would be to compare the age-specific population immunity and incidence that would be estimated using a catalytic model with the other model inputs/outputs (FOI and rate of immune waning). How does this compare to estimates of the FOI from a catalytic model fit to the age-specific case data?

As an alternative to high levels of complete immunity, there may be varying levels of partial immunity from prior infection/exposure with new exposures leading to asymptomatic infections with immune boosting (e.g., King et al. 2008 Nature) and lower levels of shedding. It might be useful to model this possibility, with vaccine providing additional protection beyond naturally-derived immunity. In lines 372-377 you mention the duration of immunity from challenge studies, but the infection of naive individuals in these studies might not represent the infections or "incidence rate" observed in the cited Bangladesh study of vibriocidal titers.

Having a single environmental reservoir for the entire population might also overestimate its contribution to transmission. Metapopulation modeling was mentioned in the introduction, so it would be useful to understand if was considered here and if not, why not? Particularly given that data from this outbreak in Kalemie was previously used to show that risk of infection was higher within 200m of a previous case (Azman et al. 2018 JID), suggesting that the environmental risk varies spatially (or direct transmission is more common than estimated here).

From line 130 ("...northern part of the city laying pipes...") it sounds like the WASH intervention was more narrowly geographically targeted. Is this assumption correct, and if so could the model be subdivided in a way that would incorporate this difference?

Minor comments

1. It would be helpful to have a table with parameter descriptions in the main text for easy reference

2. In the Results and/or 1st paragraph of the Intro it would be helpful to also present the cases averted by vaccination and WASH as a % of total cases in the absence of either intervention. The raw numbers seem high relative to the number of observed cases, but I assume that is because you also estimate a large fraction of infections/cases were unreported?

3. While the counterfactual simulations of incidence under interventions that completely prevent contamination or exposure of the environment are nice examples of the potential power of improving access to clean water and sanitation, real-world WASH interventions rarely have impacts as large as those estimated here. It might be useful to also see the impact of interventions with reductions in contamination and/or exposure that are reflectively of some previous interventions

PLOS authors have the option to publish the peer review history of their article (what does this mean?). If published, this will include your full peer review and any attached files.

Reviewer #1: No

**Figure resubmission:**
---

## [Editor Report · Decision Letter 1]

25 Jan 2025

Dear Dr. Blake,

We are pleased to inform you that your manuscript 'Impact of a multi-pronged cholera intervention in an endemic setting' has been provisionally accepted for publication in PLOS Neglected Tropical Diseases.

Best regards,

Joseph M. Vinetz

Section Editor

Shaden Kamhawi

co-Editor-in-Chief

Paul Brindley

co-Editor-in-Chief

---

## [Editor Report · Acceptance letter]

31 Jan 2025

Dear Dr. Blake,

We are delighted to inform you that your manuscript, "Impact of a multi-pronged cholera intervention in an endemic setting," has been formally accepted for publication in PLOS Neglected Tropical Diseases.

Best regards,

Shaden Kamhawi

co-Editor-in-Chief

Paul Brindley

co-Editor-in-Chief
